# Reasons for Treatment Discontinuation and Their Effect on Outcomes of Immunotherapy in Southwest Finland: A Retrospective, Real-World Cohort Study

**DOI:** 10.3390/cancers16040709

**Published:** 2024-02-07

**Authors:** Saana Virtanen, Heidi Pihlman, Maria Silvoniemi, Pia Vihinen, Panu Jaakkola, Kalle E. Mattila

**Affiliations:** 1Department of Oncology and Radiotherapy, Fican West Cancer Centre, Turku University Hospital, University of Turku, 20521 Turku, Finland; saana.maaria.virtanen@tyks.fi (S.V.); heidi.pihlman@tyks.fi (H.P.); pia.vihinen@tyks.fi (P.V.); panu.jaakkola@tyks.fi (P.J.); 2Department of Respiratory Medicine, Turku University Hospital, University of Turku, 20521 Turku, Finland; maria.silvoniemi@tyks.fi; 3InFLAMES Research Flagship Center, University of Turku, 20521 Turku, Finland

**Keywords:** immune checkpoint inhibitors, immune-related adverse events, immunotherapy, overall survival, progression-free survival, real-world evidence, treatment discontinuation

## Abstract

**Simple Summary:**

Currently, immune checkpoint inhibitors are the backbone of treatment for multiple different types of advanced cancer in routine clinical practice. However, most patients still discontinue treatment due to disease progression, and some patients discontinue treatment after severe immune-related adverse events. We aimed to evaluate the reasons for treatment discontinuation in adult patients with advanced cancer (mainly non-small cell lung cancer, melanoma, and kidney cancer) who had received immune checkpoint inhibitors in the first-line or later treatment lines outside clinical trials in Southwest Finland. In this study, it was found that disease progression was the most common reason for treatment discontinuation (in 62% of the patients) followed by immune-related adverse events (17%) and disease control or radiological response (12%). The patients who discontinued treatment due to immune-related adverse events and disease control or radiological response had favorable treatment outcomes, and 46% of them remained alive and progression-free during follow-up.

**Abstract:**

Immune checkpoint inhibitors (ICI) have improved survival in several cancer types. Still, most patients develop disease progression during or after treatment. We evaluated the reasons for treatment discontinuation and their effect on treatment outcomes in adult patients with advanced cancer with ICI in the first or later treatment lines in Southwest Finland between 1 January 2015 and 31 December 2021. Baseline characteristics and treatment outcomes were retrospectively obtained from the electronic medical records. There were 317 patients with 15 different cancer types, most commonly non-small cell lung cancer, melanoma, and kidney cancer, treated with ICI outside clinical trials. During follow-up, 94% of the patients had discontinued treatment. A total of 62% was due to disease progression, 17% due to immune-related adverse events (irAEs), 12% after achieving disease control or radiological response, and 9% due to poor performance status. The median progression-free survival (mPFS) was 5.4 months and the median overall survival (mOS) was 20.3 months in the whole cohort. Longer mPFS and mOS were observed in patients who discontinued ICI due to irAEs (24.3 and 49.2 months) and after disease control (49.7 months and not reached). In total, 46% of the patients who discontinued ICI after irAEs or disease control remained alive and progression-free during follow-up.

## 1. Introduction

Immune checkpoint inhibitors (ICI) have become the backbone of the treatment for several different histological types of cancer during the last ten years [1]. By November 2023, the European Medicines Agency (EMA) approved one CTLA-4 inhibitor (ipilimumab), four PD-1 inhibitors (nivolumab, pembrolizumab, cemiplimab, and dostarlimab), three PD-L1 inhibitors (atezolizumab, avelumab, and durvalumab), and one LAG-3 inhibitor (relatlimab) either as monotherapy or in combination with other ICI, chemotherapy, or targeted therapy [2]. Although the five-year overall survival (OS) rates reached 13% to 39% with PD-1/L1 blockade alone, the five-year progression-free survival (PFS) rates have remained 8% to 28% in patients with advanced non-small cell lung cancer (NSCLC), renal cell carcinoma (RCC), and melanoma in clinical trials [3,4,5], and the treatment is commonly discontinued due to disease progression. Biomarker-guided selection of patients, e.g., based on tumor cell PD-L1 expression in NSCLC [6] or microsatellite-instability-high (MSI-high) tumor type in colorectal cancer [7], has managed to enrich patients who gain long-term benefit from ICI in some tumor types. Yet, patients eligible for immunotherapy are offered ICI without any predictive biomarkers in several indications.

Besides disease progression, some patients who receive ICI will develop immune-related adverse events (irAEs), leading to the discontinuation of treatment. Usually, irAEs resolve with corticosteroids, but it might not be feasible to resume immunotherapy after severe irAEs, and there have also been lethal cases of, e.g., myocarditis or encephalitis [8]. In a Dutch register study, 17% of patients who had predominantly received PD-1/L1 inhibitors experienced grade ≥ 3 irAEs, but only 28% of them had to discontinue ICI permanently [9]. There is evidence of an improved benefit from immunotherapy in patients who develop irAEs [10,11,12], although conflicting results have also been reported [13]. Currently, PD-1/L1 blockade is frequently combined with CTLA-4 inhibitors, chemotherapy, or receptor tyrosine kinase inhibitors (TKI), and combination therapies have managed to improve treatment outcomes [14,15,16,17,18]. However, more than 50% of patients have experienced grade ≥ 3 adverse events with combination therapies [14,15,16,17,18]. Therefore, regular monitoring and prompt management of irAEs are required during immunotherapy.

The population of patients treated in routine clinical practice is more heterogeneous compared to patients treated in clinical trials. In routine clinical practice, there are often patients with brain metastases, impaired performance status, and significant comorbidities including autoimmune diseases that might lead to early treatment discontinuation and impaired treatment outcomes. In earlier population-based studies, it has been observed that even up to 53–55% of patients with advanced melanoma were not “trial-like” [19,20]. In this retrospective, real-world study, we aimed to evaluate the reasons for treatment discontinuation and their effect on treatment outcomes in a cohort of adult patients with advanced cancer who had received ICI in routine clinical practice in Southwest Finland. The incidence of disease progression (PD) in patients who discontinue ICI after irAEs, disease control, or radiological response was also studied.

## 2. Materials and Methods

This was a retrospective, register study utilizing data obtained from the electronic medical records of the Wellbeing Services County of Southwest Finland (WSCSF), which provides specialized healthcare services for 490,000 inhabitants. In the WSCSF, the treatment of adult patients with cancer is centralized at Turku University Hospital. All patients aged ≥18 years who had received ipilimumab, nivolumab, pembrolizumab, cemiplimab, atezolizumab, avelumab, durvalumab, or relatlimab between 1 January 2015 and 31 December 2021 were identified from the hospital pharmacy system. The investigators (S.V., H.P., and K.E.M.) reviewed the electronic medical records of the identified patients and manually completed the study database. Overall, 384 adult patients with cancer received ICI during the study period. After excluding 35 patients with neoadjuvant or adjuvant treatment for resectable localized cancer and 32 patients treated in clinical trials, a total of 317 adult patients with inoperable locally advanced or metastatic cancer were included in the final study cohort. The primary objectives were to evaluate the reasons for treatment discontinuation and to study their effect on PFS and OS in real-world patients treated with ICI outside clinical trials.

Histological tumor type and patient characteristics were obtained from the electronic medical records. Baseline characteristics included age, sex, ECOG performance status, and the presence of brain metastases at the index date, which was the date of the first ICI infusion. Information on the treatment line, treatment regimen, the number of infusions received, the duration of treatment, and the reasons for treatment discontinuation were collected. The reasons for treatment discontinuation were classified as radiologically or clinically confirmed PD, poor performance status, immune-related adverse events (irAEs), and disease control or radiological response achieved with the treatment. Ongoing treatments were also registered. We did not determine the objective response rate because radiological response evaluations were not performed according to RECIST 1.1. [21] nor iRECIST criteria [22] in routine clinical practice. The date of radiological or clinical PD was determined by the study investigators (S.V., H.P., and K.E.M.), and the date of death or the last follow-up visit was also obtained. The survival follow-up ended on 1 November 2022.

Categorical variables were presented with numbers and percentages and continuous variables with median and range. Categorical variables were compared using Pearson’s two-sided Chi-Square test. Kaplan–Meier method was used to estimate PFS and OS. PFS was determined as the time from the index date to the date of PD (event), death (event), or the last follow-up visit (censored). OS was determined as the time from the index date to the date of death (event) or the last follow-up visit (censored). The median OS (mOS) and the median PFS (mPFS) estimates were presented with 95% confidence intervals (95% CI). Univariate hazard ratios (HR) with 95% CI and the Log Rank test were used to analyze differences between patient groups. Statistical analyses were performed using IBM SPSS Statistics version 27.

The study was approved by the Institutional Review Board of Turku University Hospital (License Number T92/2022). Informed consent was waived due to the retrospective design of this study according to the Finnish legislation on the secondary use of health data. Data were anonymized before statistical analyses and handled in a manner that met general regulations on data protection.

## 3. Results

### 3.1. Patient and Treatment Characteristics

There were patients with 15 different histological types of cancer treated with ICI in routine clinical practice in Southwest Finland (Appendix A). Baseline characteristics and the treatment of study patients are described in Table 1: the most common cancer types were NSCLC, cutaneous melanoma, and RCC. The median age was 67 (27–90) years and there was a male predominance resulting from the large proportion of patients with NSCLC in the study cohort. Only 8% of the patients treated with ICI had markedly impaired performance status (ECOG ≥ 2), and 7% of patients had baseline brain metastases. In this study cohort, 276 patients (87%) received PD-1/L1 inhibitor monotherapy, 1 patient (0.3%) received ipilimumab monotherapy, and 40 patients (13%) received combination therapies. A total of 141 (44%) patients received ICI as the first-line treatment and 176 (56%) in later treatment lines. The median duration of treatment was 2.8 (0.5–37.5) months in the whole study cohort. In total, 34 patients (11%) continued treatment beyond one year and 8 patients (3%) beyond two years.

### 3.2. Reasons for Treatment Discontinuation

There were 20 (6%) patients who had ongoing treatment at the end of the follow-up. Among 297 patients who had discontinued ICI, PD was the most common reason for treatment discontinuation occurring in 185 patients (62% of the patients who discontinued ICI). A total of 50 patients (17%) discontinued ICI after developing irAEs. Another 35 patients (12%) discontinued ICI after achieving disease control or radiological response to treatment. In addition, 27 patients (9%) were ineligible for further treatment because of poor performance status.

Although there were significant differences in the treatment regimens used (PD-1/L1 inhibitor monotherapy, CTLA-4 inhibitor monotherapy, CTLA-4 + PD-1 inhibitor combination therapy, or PD-1/L1 inhibitor + chemotherapy) in patients with ICI in the first-line or later treatment lines (*p* < 0.001), the reasons for treatment discontinuation were similar regardless of the treatment line (*p* = 0.49) (Appendix A).

All irAEs leading to treatment discontinuation are described in Appendix A. The most common irAE leading to treatment discontinuation was pneumonitis with an incidence of 4.1% in the whole study cohort followed by hepatitis (3.4%) and colitis (3.1%). The median duration of treatment was only 1.4 (0.5–35.9) months in patients who discontinued ICI due to irAEs. IrAEs leading to treatment discontinuation were mostly grade 3–4 (in 31 patients, 62% of the patients discontinued treatment due to irAEs), but there were also 18 patients (36%) with intolerable or recurrent grade 2 irAEs and 1 patient (2%) with grade 5 encephalitis leading to treatment discontinuation in this study cohort (Appendix A). Among the patients who discontinued treatment due to irAEs, there were no significant associations with the grade of irAEs and PFS (Log rank *p* = 0.24) or OS (Log rank *p* = 0.18).

### 3.3. Outcomes of Immunotherapy

With the median follow-up of 28.6 months, the investigator’s assessed mPFS was 5.4 (4.5–6.3) months and the mOS 20.3 (15.4–25.2) months in the whole study cohort. The mPFS and mOS were 7.1 (5.0–9.2) months and 37.6 (24.1–51.1) months in patients with ICI used in the first-line compared to 4.5 (3.7–5.2) months and 16.0 (13.8–18.2) months in patients with ICI used in the later treatment lines (*p* = 0.004 and *p* = 0.003, respectively). The mPFS and mOS were 4.9 (3.9–5.9) months and 20.3 (15.1–25.5) months with PD-1/L1 inhibitor monotherapy compared to 6.9 (4.7–9.1) months and not reached with other regimens (*p* = 0.54 and *p* = 0.99, respectively).

When patients were stratified according to cancer types, the mPFS and mOS results were 4.9 (4.0–5.7) months and 16.0 (13.2–18.9) months in patients with NSCLC, 5.6 (3.3–7.9) months and 35.2 (16.1–54.4) months in patients with cutaneous melanoma, 6.5 (4.4–8.6) months and 41.7 (21.7–61.7) months in patients with RCC, 3.2 (0–6.5) months and 11.9 (6.6–17.3) months in patients with urothelial cancer, and 10.9 (2.4–19.4) months and not reached in other cancer types, respectively (Figure 1).

Among 297 patients who discontinued ICI during the study follow-up, the mPFS was 4.8 (4.1–5.5) months and the mOS 18.5 (14.3–22.7) months. Treatment outcomes according to the reason for treatment discontinuation are described in Figure 2: The mPFS was 49.7 months (N/A) and the mOS was not reached in patients who discontinued ICI after achieving disease control or radiological response. Long PFS and OS were also observed in patients who discontinued ICI after developing irAEs (mPFS 24.3 (15.3–33.3) months and mOS 49.2 months (N/A)). In contrast, the mPFS was 3.1 (2.2–4.1) months and the mOS 14.1 (12.6–15.5) months in patients who discontinued ICI due to PD and only 2.1 (1.8–2.4) months and 6.9 (0–16.0) months in patients who discontinued ICI due to poor performance status (Figure 2). The univariate HR of PD or death was 0.17 (0.099–0.30) and the univariate HR of death was 0.11 (0.047–0.28), for patients who discontinued treatment after disease control or radiological response compared to other patients. The univariate HR of PD or death was 0.38 (0.26–0.55) and the univariate HR of death was 0.32 (0.20–0.55) for patients who discontinued treatment due to irAEs compared to other patients.

The reason for treatment discontinuation had a similar effect on PFS and OS results in patients with NSCLC and in patients with other tumor types. Patients who discontinued treatment after achieving disease control or radiological response and due to irAEs had better treatment outcomes in both groups (Appendix A).

### 3.4. Disease Progression after Treatment Discontinuation

The proportion of patients who were alive and progression-free at the end of the follow-up was 36% (*n* = 18) in patients who had discontinued ICI due to irAEs and 60% (*n* = 21) in patients who had discontinued ICI after achieving disease control or radiological response. There were seven patients with radiological CR according to the study investigators, and five of them remained alive and progression-free during follow-up.

Progression-free survival status and the duration of treatment in patients who discontinued treatment due to irAEs and after achieving disease control or radiological response are shown in Figure 3. Of these two patient groups, 35 patients (41%) received ICI longer than six months and 50 patients (59%) received treatment less than six months. The proportion of patients who were alive and progression-free was 66% (*n* = 23) in patients who received ICI longer than six months compared to 32% (*n* = 16) in patients with the duration of treatment less than six months (*p* = 0.002). There were patients with long periods of time without any systemic treatment after the discontinuation of ICI (Figure 3).

## 4. Discussion

Currently, ICI is the standard treatment of multiple different cancer types either alone or in different combinations [2,3,4,5,6,7,14,15,16,17,18]. Despite some clinically applicable predictive biomarkers (e.g., tumor PD-L1 expression, MSI-high status, or high tumor mutational burden), the individual treatment outcome remains unpredictable [23]. In this real-world cohort of adult patients who received predominantly PD-1/L1 inhibitors for the treatment of advanced cancer, it was observed that patients who received ICI in routine clinical practice in Southwest Finland were mostly “trial-like”: only 8% had ECOG performance status ≥ 2 and 7% had baseline brain metastases. Nevertheless, most patients (62%) discontinued treatment due to PD with the mPFS of 5.4 months highlighting the fact that primary and secondary resistance to immunotherapy is a common clinical problem. In this study, there were 93 patients with PD less than two months from the first ICI infusion (50% of patients with PD), and some of these patients might have had a pattern of hyperprogression [24]. However, the mOS was 20.3 months in the whole study cohort due to effective subsequent treatment options utilized in 52% of the patients after immunotherapy. The shortest survival (mOS 6.9 months) was observed in the group of patients who discontinued ICI due to poor performance status indicating that the patients with an impaired baseline performance status could benefit from early palliative intervention rather than ICI treatment.

In this study cohort, 17% of patients discontinued ICI due to irAEs and their median duration of treatment was only 1.4 months resembling the results of the Dutch real-world register cohort with a 17% rate of grade ≥ 3 irAEs and 1.9 months time to the onset of grade ≥ 3 irAEs from the initiation of immunotherapy [9]. The rate of treatment discontinuation due to irAEs was higher in our real-world study cohort compared to clinical trials including selected patients with advanced melanoma, RCC, and NSCLC treated with PD-1 inhibitor monotherapy (17% vs. 9–13.6%) [4,5,6]. Regular monitoring of laboratory tests and the use of electronic patient-reported outcome applications are warranted to detect the signs and symptoms of irAEs early. There are also ongoing studies addressing the need to prevent irAEs by using prophylactic immunosuppressive agents during immunotherapy [25,26]. However, the patients who developed irAEs leading to treatment discontinuation had favorable oncologic outcomes in our study cohort as also observed earlier in clinical trials and real-world studies [3,10,11,12,18,27]. In a retrospective, real-world study of patients who received pembrolizumab for advanced NSCLC with a PFS of at least two years, 19.4% of patients had discontinued treatment early due to AEs [27]. These patients had long-lasting PFS (a minimum of 30 months [27]) which is in line with the results observed in our study. Still, only 36% of the patients who discontinued treatment due to irAEs in our study cohort remained alive and progression-free during follow-up, thereby highlighting the need for regular radiological and clinical monitoring for the signs and symptoms of PD after treatment discontinuation.

The proportion of patients who discontinued ICI after achieving disease control or radiological response was 12% in this study cohort. These patients sustained benefits from ICI with the HR of PD or death of 0.17 compared to other patients. During follow-up, 60% of these patients stayed alive and progression-free. In Finland, it is generally recommended to discontinue ICI after radiological CR is confirmed in two subsequent response evaluations or after 24 months of treatment. However, during earlier years of the study, there were recommendations to discontinue treatment earlier which led to impaired treatment outcomes [28,29]. The greater risk of PD in patients with the duration of treatment less than six months was also observed in our study population. A long duration of treatment was uncommon in Southwest Finland as only 11% of patients had received ICI beyond one year and 3% beyond two years. In a large cohort of patients with advanced NSCLC from the US, 7.5% of the patients were still on ICI therapy at two years and it was observed that treatment beyond two years did not improve overall survival anymore [30]. Novel methods utilizing, e.g., sequential liquid biopsies to detect circulating tumor DNA during cancer treatment could supplement radiological response evaluations [31,32] and might help to de-escalate or even discontinue treatment in patients with a complete molecular response. Hopefully, novel methods will be validated in prospective clinical trials and adopted into routine clinical practice in the near future [33].

The major limitations of this study are attributed to its retrospective single-center design. As we included all patients treated with ICI in routine clinical practice in Southwest Finland, there were several different cancer types with a limited number of patients who mostly received PD-1/L1 inhibitor monotherapy in the study cohort. There were different prognostic factors and treatment options available for patients with different cancer types affecting treatment outcomes and adverse events observed in this study. The results of this study were driven by the population of patients with NSCLC representing 59% of the study patients. However, the effect of the reason for treatment discontinuation on PFS and OS was similar in patients with NSCLC and in patients with other tumor types. This is in line with the results of earlier clinical trials and real-world analyses in patients with advanced NSCLC and melanoma shoving better treatment outcomes in patients who discontinue treatment early due to toxicity [3,10,11,12,18,27]. Moreover, there might be selection bias affecting the study cohort as patients with the poorest prognostic characteristics were likely deemed ineligible for immunotherapy and patients with favorable prognostic characteristics were more likely to enroll in available clinical trials during the study period. The increasing use of combination therapies will increase the rate of treatment-related adverse events and the objective response rate of ICI, affecting reasons for treatment discontinuation and hopefully improving survival outcomes. Therefore, the results of this study reflect the general outcomes of patients treated with PD-1/L1 blockade in routine clinical practice and are not directly applicable to a certain cancer type or treatment line. Potential differences between patient populations should be considered when comparing the results of different real-world studies.

## 5. Conclusions

In conclusion, 17% of adult patients with advanced cancer who received PD-1/L1 inhibitor-based therapy in routine clinical practice discontinued treatment due to irAEs, and 12% discontinued treatment after achieving disease control or radiological response, and these patients had favorable treatment outcomes. IrAEs and PD leading to treatment discontinuation occurred typically during the first months of treatment. Regular monitoring of adverse events and response to treatment is important to manage irAEs and to detect PD early while patients are still fit for subsequent treatment options. Therefore, our future research work will focus on clinically applicable biomarkers for monitoring adverse events and response to treatment.

## Figures and Tables

**Figure 1 cancers-16-00709-f001:**
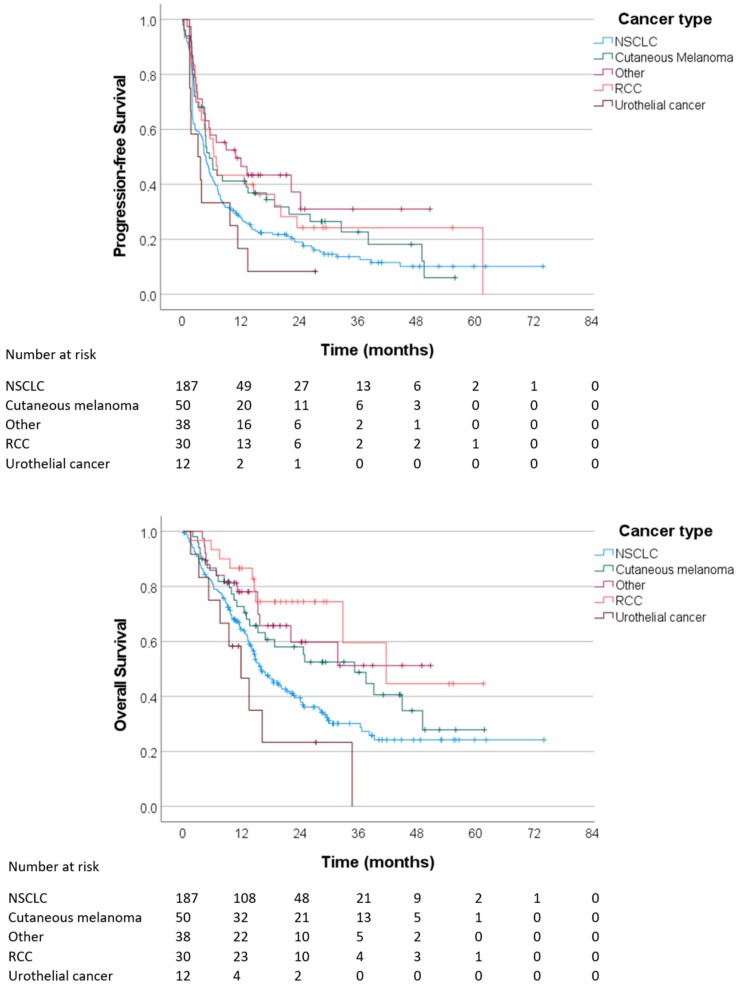
Progression-free survival and overall survival in patients treated with immune checkpoint inhibitors outside clinical trials in Southwest Finland.

**Figure 2 cancers-16-00709-f002:**
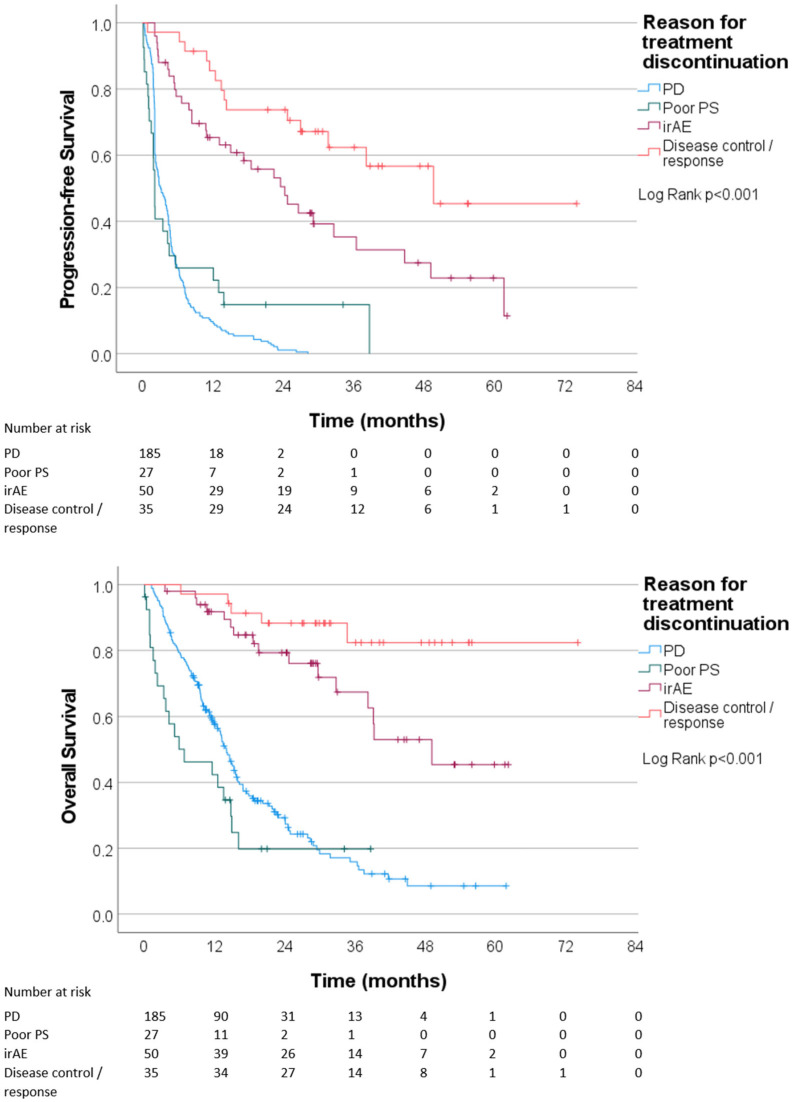
Progression-free survival and overall survival according to the reason for treatment discontinuation: disease progression (PD), poor performance status (poor PS), immune-related adverse events (irAE), and disease control or radiological response.

**Figure 3 cancers-16-00709-f003:**
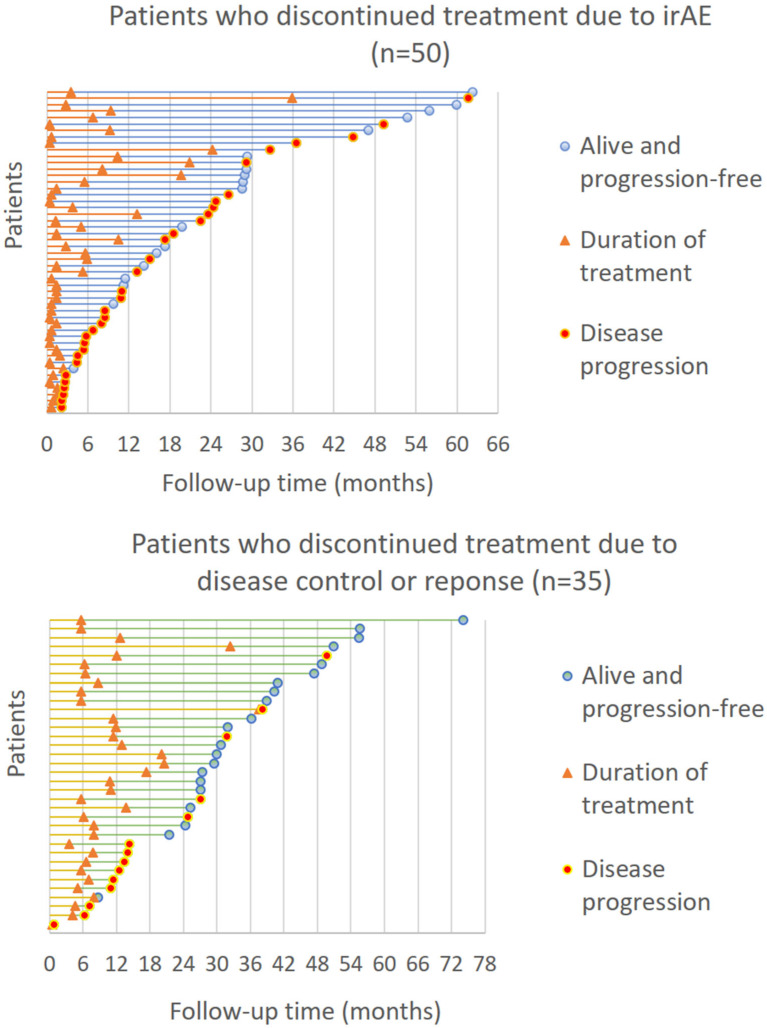
Progression-free survival status and the duration of treatment in patients who discontinued treatment due to irAEs or after achieving disease control or radiological response.

**Table 1 cancers-16-00709-t001:** Baseline characteristics and the treatment of 317 patients who received ICI for the treatment of advanced cancer in routine clinical practice in Southwest Finland.

Variable	Result
Median age	67 (27–90) years
Sex: Male	197 (62%)
Sex: Female	120 (38%)
ECOG performance status: 0	88 (28%)
ECOG performance status: 1	203 (64%)
ECOG performance status: ≥2	26 (8%)
Cancer type: Non-small cell lung cancer	187 (59%)
Cancer type: Cutaneous melanoma	50 (16%)
Cancer type: Renal cell carcinoma	30 (9%)
Cancer type: Urothelial cancer	12 (4%)
Cancer type: Other	38 (12%)
Brain metastases prior to ICI: not present	296 (93%)
Brain metastases prior to ICI: present	21 (7%)
Treatment type: PD-1/L1 inhibitor	276 (87%)
Treatment type: CTLA-4 + PD-1 inhibitor	13 (4%)
Treatment type: PD-1/L1 + chemotherapy	27 (9%)
Treatment type: CTLA-4 inhibitor	1 (0.3%)
Treatment line: First-line	141 (44%)
Treatment line: Later line	176 (56%)
Median duration of treatment	2.8 (0.5–37.5) months
Duration of treatment: <6 months	240 (76%)
Duration of treatment: 6–12 months	43 (14%)
Duration of treatment: >12 months	34 (11%)
Received systemic cancer treatment after ICI	165 (52%)

Continuous variables are presented with median (range) and categorical variables with number (percentage).

## Data Availability

Data are available upon reasonable request to the corresponding author.

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
