# Peer review of "Reasons for Treatment Discontinuation and Their Effect on Outcomes of Immunotherapy in Southwest Finland: A Retrospective, Real-World Cohort Study"

_cancers, 2024, doi:10.3390/cancers16040709_

Round 1
Reviewer 1 Report
Comments and Suggestions for Authors
This is a well presented retrospective study examining the ICIs discontinuation patterns in a part of Finland. The paper provides the relevant data and the discussion confirms the already known patterns seen previously and from the registrational clinical trials.
There is no mention about the patients for whom there is suspected hyper- progressive disease. Because it remains a relevant, yet rare event, the addition of relevant data (if any), would be of added value to this paper.
Reviewer 2 Report
Comments and Suggestions for Authors
Interesting article from Saana Virtanen et al. are describing an article regarding the outcome and the efficacy of IO treatment for patients diagnosed mainly non-small cell lung cancer, melanoma, and kidney cancer that discontinued their treatment regarding immune-related adverse events –which were almost 17% of patients. As it is showing that these group of patients had kind of a benefit of these adverse events, as the results showing that the patients who discontinued treatment due to immune related adverse events and disease control or radiological response had favorable treatment outcomes, and 46% of them remained alive and progression-free during follow-up.
These may increase the horizons for us as oncologist that also patients with severe AE or after archiving a significant response.
I have 2 minor comments:
1- Regarding to Table S2. Immune related adverse events (irAEs) leading to treatment discontinuation- please add also the grade of these AE! I'm not sure that also grades 1 or 2 AE had also good outcomes.
2- Another article regarding treatment discontinuation was published also from Israel regarding lung cancer during treating with IO- it was mentioned that 19.4% of the cohort patients, had to stop treatment early due to severe treatment-related adverse events. They all showed a prolonged response with a minimum PFS of 30 months. Please add it to the discussion. Comparing with your results which are close (your AE were 17%) and the good outcome.
Shalata W, Zolnoorian J, Migliozzi G, Jama AA, Dudnik Y, Cohen AY, Meirovitz A, Yakobson A. Long-Lasting Therapeutic Response following Treatment with Pembrolizumab in Patients with Non-Small Cell Lung Cancer: A Real-World Experience. Int J Mol Sci. 2023 Mar 21;24(6):5938. doi: 10.3390/ijms24065938. PMID: 36983011; PMCID: PMC10056863.
Reviewer 3 Report
Comments and Suggestions for Authors
I congratulate Saana Virtanen et al, for this cohort study which complements the results of the clinical studies.
The paper is very well written, it is missing a few points which would further clarify the results of the study, and its significance in comparison to the that of clinical trials. I request the addition of some points for clarification purposes and others to provide more information to the readers.
· Please clarify in the simple summary and in the abstract that your study included patients who received ICI in the first line and later settings. It is mentioned later but better to make it clear from the beginning.
· 71-72: (There is evidence of an improved benefit from immunotherapy in patients who developed irAEs). Is there a trend regarding the grade of irAE and survival i.e. higher grades are associated with better or worse survival. Is this benefit for ICI monotherapy or seen also in the combination? I suggest adding a couple of sentences to clarify this.
· You have presented the mPFS and mOS for the whole cohort, what about individual groups, especially NSCLC which represents 59% of the population?
· I agree with the authors that the study provides very important information regarding the management of patients treated in routine practice who are unlikely to be included in clinical studies.
Accordingly, it would strengthen the paper if the discussion provided general guidance/information about how the results achieved in the group’s practice compared to the results with the clinical studies (with the caveat mentioned here about different patient population, PS, etc.). A few sentences highlighting if the direction of benefit is the same. Likewise, if the types and grades of AEs seen in the cohort and clinical trials are comparable.
Addressing these points will provide the oncologists with an important piece of information on what to expect from the treatment with ICI in real life situations.
Reviewer 4 Report
Comments and Suggestions for Authors
The authors describe the experience with checkpoint inhibitors in Southwest Finland. They concluded that the most common cause of discontinuation was progressive disease, and irAE happened in around 20% of patients.
One of the major flaws of this study is that the data set is extremely heterogeneous, and the results are uninterpretable. The statistics are impressive, but it is a very heterogeneous group of patients, so it will be impossible to generalize these results.
One major improvement could be stratifying the results according to major cancer types and discussing them individually. However, given the low numbers, it may not be feasible.
Comments on the Quality of English LanguageThe English quality is good. We may need to improve with the removal of redundant statements.
